# Cardiovascular Magnetic Resonance Imaging Findings in Africans with Idiopathic Dilated Cardiomyopathy

**DOI:** 10.3390/diagnostics13040617

**Published:** 2023-02-08

**Authors:** Nqoba Tsabedze, Andre du Plessis, Dineo Mpanya, Anelia Vorster, Quinn Wells, Leonie Scholtz, Pravin Manga

**Affiliations:** 1Division of Cardiology, Department of Internal Medicine, School of Clinical Medicine, Faculty of Health Sciences, University of the Witwatersrand, Johannesburg 2193, South Africa; 2Diagnostic Radiology, Midstream Mediclinic, Centurion 1692, South Africa; 3Division of Cardiovascular Medicine, Department of Medicine, Vanderbilt University Medical Center, Nashville, TN 37232, USA

**Keywords:** idiopathic dilated cardiomyopathy, magnetic resonance imaging, cardiovascular, late gadolinium enhancement, all-cause mortality

## Abstract

In sub-Saharan Africa, idiopathic dilated cardiomyopathy (IDCM) is a common yet poorly investigated cause of heart failure. Cardiovascular magnetic resonance (CMR) imaging is the gold standard for tissue characterisation and volumetric quantification. In this paper, we present CMR findings obtained from a cohort of patients with IDCM in Southern Africa suspected of having a genetic cause of cardiomyopathy. A total of 78 IDCM study participants were referred for CMR imaging. The participants had a median left ventricular ejection fraction of 24% [interquartile range, (IQR): 18–34]. Late gadolinium enhancement (LGE) was visualised in 43 (55.1%) participants and localised in the midwall in 28 (65.0%) participants. At the time of enrolment into the study, non-survivors had a higher median left ventricular end diastolic wall mass index of 89.4 g/m^2^ (IQR: 74.5–100.6) vs. 73.6 g/m^2^ (IQR: 51.9–84.7), *p* = 0.025 and a higher median right ventricular end-systolic volume index of 86 mL/m^2^ (IQR:74–105) vs. 41 mL/m^2^ (IQR: 30–71), *p* < 0.001. After one year, 14 participants (17.9%) died. The hazard ratio for the risk of death in patients with evidence of LGE from CMR imaging was 0.435 (95% CI: 0.259–0.731; *p* = 0.002). Midwall enhancement was the most common pattern, visualised in 65% of participants. Prospective, adequately powered, and multi-centre studies across sub-Saharan Africa are required to determine the prognostic significance of CMR imaging parameters such as late gadolinium enhancement, extracellular volume fraction, and strain patterns in an African IDCM cohort.

## 1. Introduction

Idiopathic dilated cardiomyopathy (IDCM) is an endemic primary myocardial disease in sub-Saharan Africa (SSA) [1,2]. It manifests clinically with ventricular dilatation and myocardial dysfunction without obstructive coronary artery disease (CAD) or abnormal loading conditions such as hypertension and valvular heart disease [3]. The clinical management of patients with IDCM includes identifying and treating reversible causes of myocardial dysfunction, improving survival, slowing disease progression, and alleviating symptoms [4]. Therefore, excluding secondary causes of dilated cardiomyopathy (DCM) is recommended, including genetic causes in all patients with unexplained ventricular dilatation and dysfunction. However, despite a comprehensive clinical workup, in regions where genetic testing is not widely available, patients without an identifiable cause for DCM are provided a working diagnosis of IDCM.

Cardiovascular magnetic resonance (CMR) imaging is considered the gold standard for anatomical and tissue characterisation and evaluating the extent of cardiac dysfunction. Although not widely accessible in most centres across Africa, cardiac magnetic resonance imaging (MRI) plays a significant role in excluding infiltrative macrovascular and microvascular disease and defining the presence of fibrosis in patients with DCM [5]. We report CMR imaging findings in a meticulously phenotyped cohort of black Africans with a clinical working diagnosis of IDCM. The cohort described in this paper was prospectively recruited in a study designed to identify the genetic causes of myocardial dysfunction in these patients.

## 2. Materials and Methods

### 2.1. Study Design and Participants

Between July 2015 and December 2018, we screened 161 participants for idiopathic dilated cardiomyopathy. The study’s participants were identified from the inpatient cardiology wards and outpatient heart failure with reduced ejection fraction (HFrEF) clinic at the Charlotte Maxeke Johannesburg Academic Hospital (CMJAH), a quaternary referral centre in Johannesburg, South Africa. The study inclusion criteria included adults 18 years and older with a left ventricular ejection fraction (LVEF) less than or equal to 40%. The exclusion criteria included organic valvular heart disease, hypertension, coronary artery disease, human immunodeficiency virus infection, myocarditis, infiltrative disease, and metabolic conditions. In addition, patients presenting with DCM in the peripartum period, post-chemotherapy, or post-radiation therapy, were also excluded.

A detailed clinical history was obtained from all participants. This included their age at the time of the index heart failure diagnosis, a family history of sudden cardiac death, comorbidities, and heart failure symptoms. In addition, a physical examination, which focused on identifying signs of heart failure, was performed on all participants. During recruitment, screening laboratory biochemical tests were performed on all study participants to exclude secondary causes of a DCM. These included haemoglobin, platelet count, C-reactive protein, a lipogram, thyroid and renal function tests, cardiac biomarkers, and micro-nutrient serum levels. A twelve-lead electrocardiogram (ECG) and a 2D transthoracic echocardiogram (General Electric Vivid 9 4D) were performed on all recruited participants. In addition, a diagnostic coronary angiogram was performed to exclude obstructive coronary artery disease.

All study participants received these investigations, including CMR imaging, performed within 72 h of recruitment. In this study, IDCM was defined as left ventricular dysfunction (LVEF ≤ 40%) in the absence of CAD (normal diagnostic angiography), valvular heart diseases, infiltrative disease, and metabolic abnormalities. The study complied with the Declaration of Helsinki and informed consent was obtained from the study participants. Approval to conduct the study was granted by the University of the Witwatersrand Human Research Ethics Committee (certificate number: M150467). Informed consent was obtained from all study participants.

### 2.2. Cardiac Magnetic Resonance Imaging Protocol and Image Analysis

A 1.5 T whole-body scanner (Philips) with an ECG triggering device was used for CMR imaging. Respiratory bellows were placed on the patient’s abdomen throughout the imaging process. After the acquisition of localisation images, continuous short-axis cine images of the left ventricle were obtained using steady state free precession sequence at end-expiration.

Ventricular volumes and left- and right-ventricular ejection fractions were calculated using four-chamber and short-axis slice summation.

Images depicting late gadolinium enhancement (LGE) were acquired approximately 15 min after administering 0.1 mmol/kg of gadobenate dimeglumine (Bracco Diagnostics Inc., Princeton, NJ, USA) at an injection rate of 2 mL per second. The presence of LGE was evaluated using segmented inversion recovery prepared for true fast imaging. The images were visually analysed for the presence and extent of LGE. Two radiologists were available for image interpretation, and a single radiologist independently reviewed each set of CMR images. The visual scoring method was based on the 17-segment model. The percentage of the myocardium with LGE was calculated by counting the number of segments with LGE and dividing by 17.

### 2.3. Patient Follow-Up and Study Endpoints

The CMJAH is a quaternary referral specialist centre where advanced cardiac patients are preferentially referred and definitively managed. Thus, outcome data were first collected from the Electronic Health Record System, which captures all patients admitted to the cardiology wards at the CMJAH. In addition, all-cause mortality, frequency of hospitalisations (in any hospital, including CMJAH), and the occurrence of thromboembolic complications were documented after a telephonic interview with the study participants or their next-of-kin after a median follow-up duration of 12 [interquartile range (IQR): 8.8–16.8] months.

### 2.4. Statistical Analysis

Categorical variables are expressed as counts and percentages and were compared for the study outcome using a Chi-square test. Continuous variables with a normal and non-normal distribution are expressed as mean and standard deviation, as well as the median and IQR, respectively. The Student’s *t*-test and the Wilcoxon rank sum (Mann–Whitney) test were used for comparing the mean and median, respectively. Confidence intervals were set at 95%, and *p* < 0.05 was regarded as statistically significant. A Cox proportional hazards model was created to assess for risk factors for all-cause mortality. Analyses were conducted using STATA version 16.0 (StataCorp, College Station, TX, USA).

## 3. Results

### 3.1. Baseline Clinical Characteristics

After excluding participants lost to follow-up (Figure 1), the final study cohort comprised 78 participants consisting of 53 (67.9%) males. The mean age was 47.3 ± 13.3 years (Table 1). In addition, there were 73 (93.6%) native Africans. The median left ventricular ejection fraction (LVEF) on CMR imaging was 24% (IQR: 18–34), and non-survivors had a lower median LVEF of 18% (IQR: 13–24) vs. 24% (IQR: 18–36), *p* = 0.042. Thirteen (16.7%) participants had no overt clinical symptoms of congestive cardiac failure in the study cohort. The remaining participants complained of dyspnea on minimal exertion (62.8%), paroxysmal nocturnal dyspnea (42.3%), orthopnea (58.9%), palpitations (35.9%), and syncope (7.7%), or a combination of these symptoms.

### 3.2. Cardiovascular Magnetic Resonance Imaging

The ventricular volumetric parameters are reported in Table 2. Non-survivors had a higher median left ventricular end diastolic wall mass index of 89.4 g/m^2^ (IQR: 74.5–100.6) vs. 73.6 g/m^2^ (IQR: 51.9–84.7), *p* = 0.025, a higher median right ventricular end-systolic volume index of 86 mL/m^2^ (IQR: 74–105) vs. 41 mL/m^2^ (IQR: 30–71), *p* < 0.001, and a higher median right ventricular end diastolic volume index 114 mL/m^2^ (IQR: 101–147) vs. 70 mL/m^2^ (IQR: 56–94), *p* < 0.001. The hazard ratio for the risk of death in patients presenting with evidence of LGE on CMR imaging was 0.435 (95% CI: 0.259–0.731; *p* = 0.002). However, the Cox proportional hazards model identified none of the clinical and CMR imaging parameters as statistically significant covariates for all-cause mortality. There were 43 (55.1%) participants with evidence of LGE on imaging. Late gadolinium enhancement was visualised in the midwall in 28 (65.0%) participants, and four (9.3%) had LGE in both the right ventricular insertion point and midwall (Table 3). In addition, three (7.0%) participants visualised a transmural LGE pattern, and two (4.6%) participants had areas of focal LGE. Midwall and subendocardial and epicardial LGE patterns are depicted in Figure 2. A total of nine patients had an LGE pattern commonly associated with ischaemic heart disease.

Late gadolinium enhancement occupied less than 50% of the left ventricle in 38 (48.7%) participants and 50–75% of the left ventricle in 5 (6.4%) participants. Late gadolinium enhancement was visualised in the basal anteroseptal, mid-inferoseptal and mid-anteroseptal segments in 85%, 70%, and 53% of participants, respectively (Figure 3). Features of left ventricular non-compaction were found in five (11.6%) participants with LGE. Furthermore, pericardial and pleural effusions were diagnosed in 23 (29.5%) and 11 (14.1%) participants in our cohort, respectively.

### 3.3. Electrocardiogram and Echocardiogram Parameters

At the time of recruitment into the study, the median PR interval did not differ significantly between survivors and non-survivors, measuring [177 (IQR: 158–203) and 192 (IQR: 169–222), *p* = 0.156] ms, respectively. In the entire cohort, 32 (41.0%) participants had a QRS duration >110 ms. The median QRS duration was 111 (IQR: 98–148) ms in participants with LGE, and study participants who showed no evidence of LGE upon CMR imaging had a median QRS duration of 97 (IQR: 88–117) ms (*p* = 0.015).

For echocardiography, the mean left atrial diameter was 43.8 ± 6.5 mm. This did not differ significantly between survivors and non-survivors (43.1 ± 6.4 vs. 46.8 ± 6.0 mm, *p* = 0.05). The mean left ventricle fractional shortening was 12.8 ± 5.7%, which differed significantly between survivors and non-survivors (13.5 ± 5.7 vs. 9.6 ± 5.0%, *p* = 0.036). The mean global longitudinal strain was −6.8 ± 4.6%, and participants that survived had a mean global longitudinal strain of −7.3 ± 4.0%. In contrast, non-survivors had a mean global longitudinal strain of −4.5 ± 6.3% (*p* = 0.046). There was no correlation between global longitudinal strain and LGE (r = 0.1173). During wall motion analysis, global hypokinesis was noted in 60 (76.9%) participants. 

### 3.4. Follow-Up and Endpoints

After a median follow-up duration of 12 (IQR: 8.8–16.8) months, 14 (17.9%) participants died. Thromboembolic complications were uncommon and were reported in only three participants. Each participant was diagnosed with a cerebrovascular accident, lower limb deep vein thrombosis, and pulmonary embolism. Nine (11.5%) study participants were hospitalised during the follow-up period.

## 4. Discussion

In this study, participants with DCM were referred for CMR imaging to further exclude primary and secondary myocardial diseases. In addition, LGE was also assessed based on its prognostic value in patients with dilated cardiomyopathy. The identification of LGE in patients with non-ischaemic cardiomyopathy is associated with an increased risk of all-cause mortality, rehospitalisation, ventricular dyssynchrony, spontaneous and inducible ventricular arrhythmias, and sudden cardiac death [6,7,8].

More than half of the participants in our study showed evidence of LGE on CMR imaging, and 65% of the study participants presented with evidence of LGE in the midwall. Nine participants had an LGE pattern suggestive of ischaemic cardiomyopathy. However, their diagnostic angiograms showed normal epicardial coronary arteries. These findings are supported by McCrohon and colleagues, who studied 90 patients with DCM and 15 control subjects. In their study, 13% of DCM patients with unobstructed coronary arteries upon angiography presented with subendocardial and transmural LGE patterns during CMR imaging [9], implying that microvascular ischaemia may be an important cause of ischaemic LGE patterns. However, in our cohort, we did not assess the presence of microvascular disease while performing the diagnostic coronary angiograms. Furthermore, we identified isolated LGE in the right ventricular insertion point in three participants. Isolated right ventricular insertion point LGE has been reported in several studies and shown not to convey a worse prognosis [10,11].

In a meta-analysis conducted by Duan et al., evaluating the prognostic value of LGE in dilated cardiomyopathy, patients presenting with LGE upon CMR imaging had a three-fold increased risk of all-cause mortality [12]. In another systematic review, the hazard ratio for all-cause mortality in non-ischaemic cardiomyopathy patients with LGE on imaging was 2.74 (95% CI: 2.0–3.74) *p* < 0.001 [6]. Moreover, a meta-analysis by Kuruvilla and colleagues reported a higher risk of all-cause mortality in patients with non-ischaemic cardiomyopathy with LGE (odd ratio = 3.27; 95% CI: 1.94–5.51; *p* < 0.00001) [7]. In our study, survival rates did not differ significantly in participants with and without LGE, probably because of the relatively small sample size. A few studies on DCM patients with smaller sample sizes have reported a lack of association between LGE and mortality. Looi et al. studied 103 patients with DCM and found evidence of LGE in 30% of patients. After a follow-up duration of 2 years, nine deaths occurred, and there was no statistically significant difference in all-cause mortality in patients with and without LGE [13]. Moreover, the odds ratio for the occurrence of major cardiovascular events (MACEs) was 0.77 (95% CI: 0.29–1.97; *p* > 0.05), and therefore was not statistically significant. In their study, Looi and colleagues defined a MACE as the incidence of death, cardiac transplant, ventricular arrhythmias, and heart failure-related hospitalisation. They attributed their study’s lack of an association between MACEs and LGE to a lower-risk patient profile. In their study, 75% of DCM patients were in NYHA class 1 and had a mean LVEF of 32 ± 12% [13]. Another study by Masci and colleagues reported a relationship between the absence of LGE and left ventricular reverse remodelling in 58 IDCM patients. Among the patients studied, eight died after two years of follow-up [14]. A higher rate of death in patients with evidence of LGE on imaging was not reported in the study by Masci et al.

Semi-quantitative and quantitative parameters generated by commercially available software for processing CMR images may be of value in improving the diagnosis and specificity of imaging findings. In our study, we used semi-quantitative indices to quantify the burden of LGE on images. This may have contributed to the lack of a linear relationship between the presence of LGE and mortality in our study. Moreover, our patients may have had diffuse fibrosis, which may be challenging to visualise in the absence of normal reference myocardium, or small microscopic fibrosis, which may be below the resolution of our scanner.

N-terminal, pro-brain natriuretic peptide (pro-BNP) levels are useful for the diagnosis of acute heart failure, the prognostication of patients with ischaemic and non-ischaemic cardiomyopathy, and for predicting the risk of cardiovascular and all-cause mortality in the general population [15,16]. In our study, pro-BNP levels were five-times higher in non-survivors, with a median of 5573 (IQR: 2471–8188) vs. 1106 (IQR: 421–2826), *p* = 0.001, suggesting that this biomarker, which is secreted in response to excessive myocyte stretching, could be a valuable marker for risk stratifying patients with IDCM.

A prolonged duration of the QRS complex, greater than 110 ms, did not independently predict mortality in our study. Unlike a study by Hombach et al., LGE in patients with idiopathic DCM who had a QRS duration >110 ms and diabetes mellitus was found to be significant predictor of cardiac death or sudden cardiac death from ventricular flutter or fibrillation [17]. Despite removing all participants with right ventricular insertion point LGE and an LGE pattern suggestive of ischaemic heart disease, none of our study’s clinical and imaging variables independently predicted mortality.

Global longitudinal strain (GLS) measurements are a useful prognostic marker, since abnormalities in strain patterns are often observed, despite preserved left ventricular function. In a study involving 15 and 33 patients with ischaemic heart disease and non-ischaemic heart disease, respectively, the median GLS measurements were −15.6 (IQR: −17.9 to −11.6) and −16.0 (IQR: −19.1 to −12.7) in patients with and without LGE on imaging, respectively (*p* = 0.212) [18]. In that study, which Erley and colleagues conducted, the global circumferential strain (GSC) had a stronger association with LGE than GLS, with an area under the receiver operating characteristic curve of 0.77–078 vs. 0.67–0.72 [18]. In our study, there was no correlation between GLS generated using speckle tracking during echocardiography and the presence of LGE during CMR imaging, probably because of the limited sample size.

Cardiac tissue characterisation using CMR imaging T1 mapping and the extracellular volume fraction in DCM patients are two of the most useful techniques for detecting interstitial fibrosis [19,20]. Late gadolinium enhancement is a valuable surrogate for replacement fibrosis in the heart, whereas T1 mapping and ECV fraction can reliably demonstrate the presence of interstitial fibrosis, a common finding in patients with DCM [20]. In fibrotic hearts, both native T1 and ECV on the pre-contrast map are prolonged, and the post-contrast map shows a shortened T1 time due to the accumulation of gadolinium in the extracellular space [21,22]. The use of dedicated software for T1 mapping and the ECV fraction in our DCM cohort may have played a significant role in the risk stratification and prognostication of our patients [23].

This study had several limitations. This was a single-centre observational study with a relatively small sample size. Therefore, this study was potentially unsuitable for determining if certain clinical variables could predict all-cause mortality in our IDCM cohort. A single radiologist analysed and reported CMR images. Adding a second radiologist and comparing the CMR imaging findings would have improved the diagnostic yield and assisted in measuring inter-observer coefficients. The intensity of LGE and the proportion of the wall thickness affected by LGE were not assessed. The study follow-up period was relatively short. Furthermore, our study had less female representation, as several females with DCM detected in the peripartum period were excluded. The CMR extracellular volume fraction and strain patterns were not analysed as our reporting software did not have these capabilities.

## 5. Conclusions

In this study of a carefully phenotyped cohort of Africans with IDCM, midwall enhancement was the most common pattern, visualised in 65% of participants. Prospective, adequately powered, multi-centre studies across sub-Saharan Africa are required to determine the prognostic significance of LGE on CMR imaging parameters such as LGE, ECV, and strain patterns in an African IDCM cohort.

## Figures and Tables

**Figure 1 diagnostics-13-00617-f001:**
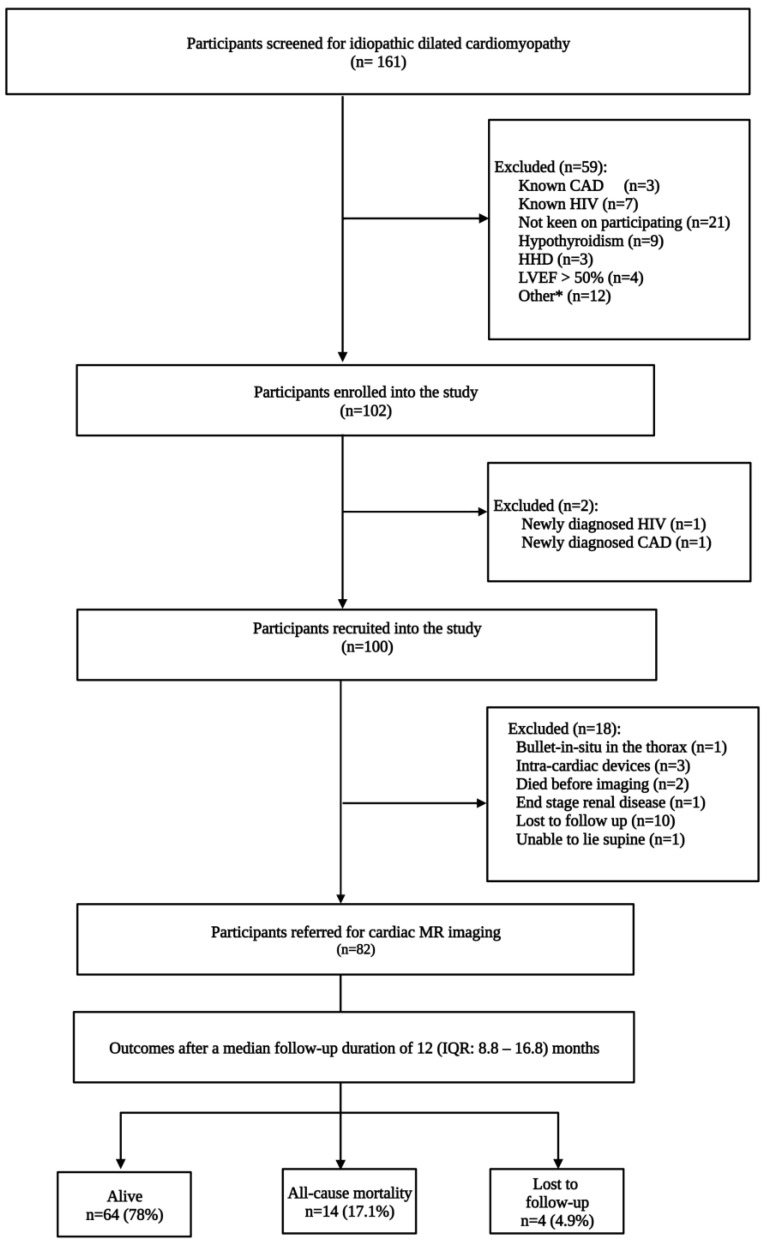
Flow chart detailing the identification of the study cohort. CAD = coronary artery disease, HHD = hypertensive heart disease, HIV = human immunodeficiency virus, LVEF = left ventricular ejection fraction. Other * refers to peripartum cardiomyopathy (*n* = 1), malignancy (*n* = 2), organic valvular heart diseases (*n* = 2), pacemaker in situ (*n* = 1), end-stage renal impairment (*n* = 3), recreational substance abuse (*n* = 1), unstable patient (*n* = 1), dilated cardiomyopathy precipitated by supraventricular tachycardia (*n* = 1).

**Figure 2 diagnostics-13-00617-f002:**
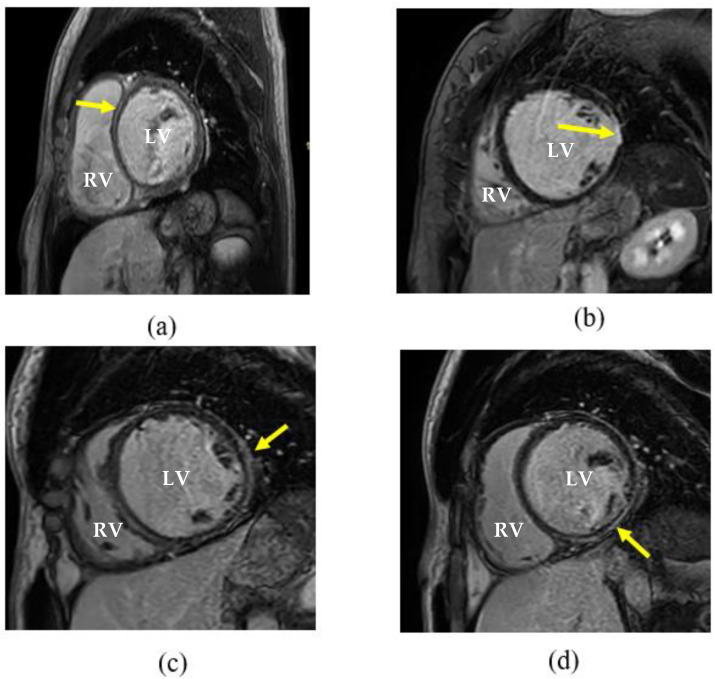
Late gadolinium enhancement cardiovascular magnetic resonance images showing the left and right ventricle. (**a**) Dilated cardiomyopathy with midwall enhancement of the septum (arrow); (**b**) subendocardial enhancement in the lateral ventricle free wall, typically associated with ischaemic heart disease; (**c**) shows an epicardial enhancement pattern in the lateral ventricle free wall and (**d**) inferior segment. LV = left ventricle; RV = right ventricle.

**Figure 3 diagnostics-13-00617-f003:**
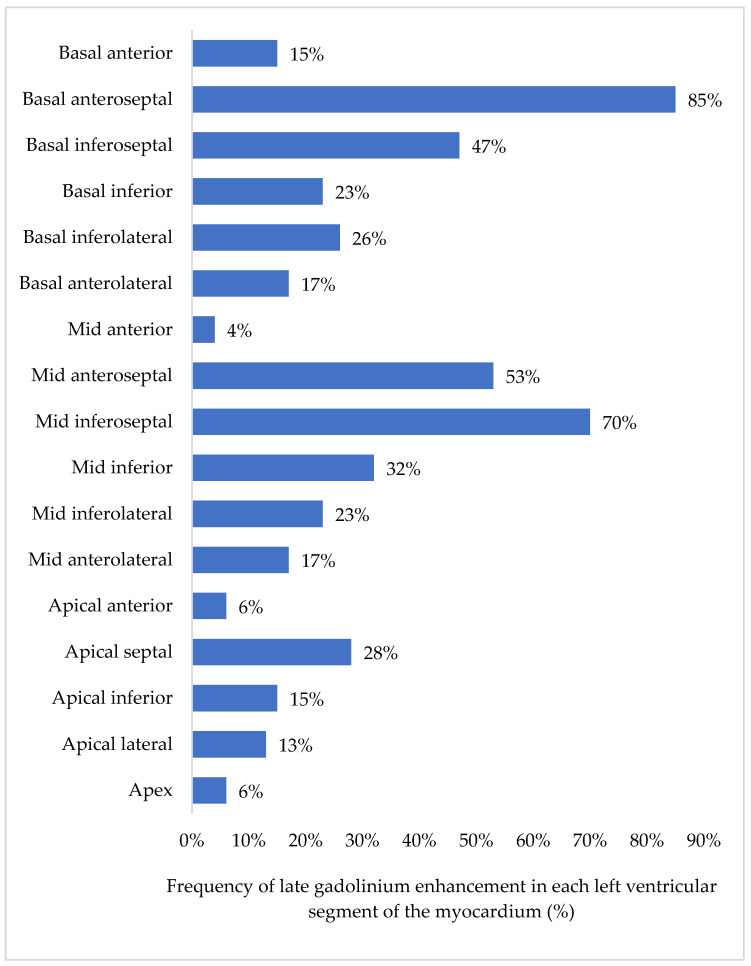
Distribution of late gadolinium enhancement in the left ventricular myocardial segments. The bars demonstrate the percentage of segments with LGE. In 85% of IDCM patients, LGE was visualised in the basal anteroseptal segment. There were 46 IDCM patients with LGE in one or more segments; therefore, the cumulative percentage will be >100%.

**Table 1 diagnostics-13-00617-t001:** Baseline clinical characteristics of patients with idiopathic dilated cardiomyopathy stratified according to one-year outcomes.

	All Patients	All-Cause Mortality	Alive	
	(*n* = 78)	(*n* = 14)	(*n* = 64)	*p*-Value
Age, years	47.3 ± 13.3	41.6 ± 14.0	48.6 ± 13.0	0.074
Male	53 (67.9)	10 (71.4)	43 (67.2)	0.758
Smoking	15 (19.2)	6 (42.9)	9 (14.1)	0.013
BMI, kg/m^2^	26.1 (23.3–30.5)	24.1 (21.4–25.3)	27.7 (23.6–31.3)	0.045
BSA, m^2^	1.9 ± 0.2	1.9 ± 0.2	1.9 ± 0.2	0.328
Heart rate, bpm	80 (69–95)	89 (69–104)	80 (69–95)	0.475
Systolic BP, mmHg	119 (101–129)	115 (99–122)	119 (106–133)	0.263
Diastolic BP, mmHg	76 (67–88)	73 (60–91)	78 (67–88)	0.442
MAP, mmHg	89 (78–101)	81 (73–93)	92 (80–101)	0.124
NYHA class				0.173
1	29 (37.2)	2 (14.3)	27 (42.2)	
2	34 (43.6)	7 (50.0)	27 (42.2)	
3	12 (15.4)	4 (28.6)	8 (12.5)	
4	3 (3.8)	1 (7.1)	2 (3.1)	
Medication				
Beta-blocker	73 (93.6)	13 (92.9)	60 (93.7)	0.837
ACE inhibitors	51 (65.4)	11 (78.6)	40 (62.5)	0.706
ARB	9 (11.5)	0 (0)	9 (14.1)	0.202
MRA	55 (70.5)	10 (71.4)	45 (70.3)	0.964
Loop diuretics	71 (91.0)	13 (92.9.1)	58 (90.6)	0.795
Statin	22 (28.2)	3 (21.4)	19 (29.7)	0.842
Sodium	140 (138–143)	140 (137–141)	140 (139–143)	0.234
Potassium	4.4 ± 0.6	4.6 ± 0.8	4.4 ± 0.5	0.167
eGFR, mL/min	70.4 (49.0–92.9)	62.6 (47.0–92.9)	75.1 (49.0–94.6)	0.610
Pro BNP	1842 (526–3860)	5573 (2471–8188)	1106 (421–2826)	0.001
Troponin I	14 (8–29)	17 (13–38)	13 (8–27)	0.234
CK-MB	1.9 (1.4–3.0)	1.6 (1.3–2.6)	2.1 (1.5–3.2)	0.181
HbA1C	6.3 (6.0–6.8)	6.4 (6.0–6.8)	6.2 (6.0–6.7)	0.768
Total cholesterol	4.1 ± 1.1	3.7 ± 0.7	4.2 ± 1.2	0.137
LDL	2.6 ± 0.9	2.1 ± 0.6	2.7 ± 0.9	0.040
HDL	1.0 (0.8–1.4)	1.0 (0.8–1.3)	1.0 (0.8–1.4)	1.000
C-reactive protein	9 (9–14)	14.5 (9.5–31)	9 (9–12)	0.005

Values are expressed as mean ± standard deviation for continuous variables with a normal distribution, or as median and (interquartile range) for continuous variables with a skewed distribution and absolute value (*n*) and percentage for categorical variables. ACE, angiotensin-converting enzyme; ARB, angiotensin receptor blocker; BMI, body mass index; BNP, beta natriuretic peptide; BP, blood pressure; BSA, body surface area; CK-MB, creatine kinase myocardial band; eGFR, estimated glomerular filtration rate; HbA1c, glycated haemoglobin; HDL, high-density lipoprotein; LDL, low-density lipoprotein; MAP, mean arterial pressure; MRA, mineralocorticoid receptor antagonist; NYHA, New York Heart Association.

**Table 2 diagnostics-13-00617-t002:** Baseline cardiovascular magnetic resonance imaging parameters stratified according to all-cause mortality.

	All Patients	All-Cause Mortality	Alive	
	(*n* = 78)	(*n* = 14)	(*n* = 64)	*p*-Value
LVEF (%)	24 (18–34)	18 (13–24)	24 (18–36)	0.042
LVEDV (mL)	222 (176–297)	282 (171–313)	220 (176–297)	0.509
LVESV (mL)	175 (121–243)	230 (156–305)	168 (115–238)	0.091
LVEDVI (mL/min)	119 (95–153)	138 (96–181)	117 (95–147)	0.482
LVESVI (mL/min)	93 (63–130)	123 (89–160)	88 (63–115)	0.101
LV stroke volume (mL)	56.3 ± 17.5	50.0 ± 15.3	57.8 ± 17.7	0.132
LV stroke volume index (mL/m^2^)	28.5 (22.8–35.7)	27.3 (24.1–36.6)	28.9 (22.9–35.5)	0.561
LVED wall mass (g)	131 (106–161)	156 (134–170)	123 (96–159)	0.036
LVED wall mass index (g/m^2^)	74.2 (53.6–88.9)	89.4 (74.5–100.6)	73.6 (51.9–84.7)	0.025
LV total mass (g)	194.3 ± 58.7	219.2 ± 70.0	188.8 ± 55.3	0.080
LV mass index (g/m^2^)	99 (85–114)	111 (90–147)	98 (82–113)	0.135
RVEDV (mL)	143 (103–201)	232 (161–316)	132 (96–178)	<0.001
RVESV (mL)	90 (58–147)	154 (127–273)	73 (55–130)	<0.001
RVEF (%)	34.2 ± 16.3	23.1 ± 12.0	36.7 ± 16.2	0.004
RVEDVI (mL/m^2^)	74 (58–109)	114 (101–147)	70 (56–94)	<0.001
RVESVI (mL/m^2^)	46 (32–82)	86 (74–105)	41 (30–71)	<0.001
RV stroke volume (mL)	48 (29–68)	50 (27–74)	48 (30–68)	0.889
RV ED wall mass (g)	139 ± 47.8	163.6 ± 8.7	135.2 ± 50.5	0.455
Cardiac output (L/min)	4.1 (3.2–5.3)	4.4 (3.2–5.5)	4.0 (3.1–5.3)	0.929
Cardiac index (L/min/m^2^)	2.5 (2.1–3.8)	2.5 (2.4–2.7)	2.4 (2.1–4.0)	0.883
Cardiac density (g/mL)	1.0 (1.0–1.0)	1.0 (1.0–1.0)	1.0 (1.0–1.0)	0.633

Values are expressed as mean ± SD for continuous variables with a normal distribution, or median and interquartile range for continuous variables with a skewed distribution and absolute value (*n*) and percentage for categorical variables. ED = end diastolic; LV = left ventricle; LVEDV = left ventricular end diastolic volume; LVEDVI = left ventricular end diastolic volume index; LVEF = left ventricular ejection fraction; LVESV = left ventricular end systolic volume; LVESVI = left ventricular end systolic volume index; RVEDV = right ventricular end diastolic volume; RVEDVI = right ventricular end diastolic volume index; RVEF = right ventricular ejection fraction; RVESV = right ventricular end systolic volume.

**Table 3 diagnostics-13-00617-t003:** Patterns of late gadolinium enhancement in IDCM patients.

	*n* = 43 (%)
Midwall	28 (65)
Right ventricular insertion point (normal variant) and midwall	4 (9.3)
Right ventricular insertion point (normal variant)	3 (7)
Transmural	3 (7)
Focal	2 (4.6)
Subendocardial	2 (4.6)
Midwall and subendocardial	1 (2.3)
Midwall, subendocardial and transmural	1 (2.3)
Midwall and epicardial	1 (2.3)
Subendocardial and transmural	1 (2.3)

Values are expressed as absolute numbers and percentages.

## Data Availability

Data are unavailable due to ethical restrictions.

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
