# Peer review of "Cardiovascular Magnetic Resonance Imaging Findings in Africans with Idiopathic Dilated Cardiomyopathy"

_diagnostics, 2023, doi:10.3390/diagnostics13040617_

Round 1
Reviewer 1 Report
The aim of Tsabedze et al. was to evaluate the prognostic value of different CMR parameters in patients suspected to have a genetic cause of DCM. They enrolled 78 patients with idiopathic DCM who underwent CMR and found that there was a difference in terms of LVEF, LV mass index, RV volumes, mass index and RVEF between survivors and non survivors; however, none of the CMR parameters were independent predictors of all-cause mortality.
The manuscript is properly structured and well written. However, the manuscript would benefit by expanding the novel aspects and results of this study in a bit more detail.
Major points:
1. As the authors point out, there are several other studies that have also looked into using CMR parameters for predicting mortality in DCM. The general statement given by the authors in lines 242 and 243, that this study has limited patient numbers is not convincing, since the number of patients in the other studies is also comparable or sometimes less than this study:
- Looi et al found increased rates of MACE in 104 patients with NIDCM and LGE (HR = 0.77, 95% CI 0.3-2.0) at 2 years (Looi JL, Edwards C, Armstrong GP, Scott A, Patel H, Hart H, Christiansen JP. Characteristics and prognostic importance of myocardial fibrosis in patients with dilated cardiomyopathy assessed by contrast-enhanced cardiac magnetic resonance imaging. Clin Med Insights Cardiol. 2010 Dec 15;4:129-34. doi: 10.4137/CMC.S5900.)
- Masci et al found that myocardial LGE is a strong predictor of worse clinical outcome in 125 patients with NIDCM (hazard ratio: 5.32, 95% confidence intervals 1.60 to 17.63, p=0.006) (Masci PG, Barison A, Aquaro GD, Pingitore A, Mariotti R, Balbarini A, Passino C, Lombardi M, Emdin M. Myocardial delayed enhancement in paucisymptomatic nonischemic dilated cardiomyopathy. Int J Cardiol. 2012 May 17;157(1):43-7. doi: 10.1016/j.ijcard.2010.11.005.)
A more detailed explanation/discussion needs to be given for the varying results/parameters reported by the other groups.
2. Major adverse cardiac events should be defined in the study in addition to all cause mortality and the frequency of hospitalizations, and the prognostic significance of LGE should be assessed in relation to both of these. This study should only be viewed as descriptive if the association between CMR parameters and adverse outcomes is not investigated in more detail.
3. A power analysis of the number of patients must be performed in order to determine the effectivity and value of the conclusions made. Were there any differences regarding LGE between groups? LGE was quantified visually as the authors state in line 96-97 (“The percentage of the myocardium with LGE was calculated by counting the number of segments with LGE and dividing by 17”). However, LGE may also require quantitative analysis; thus, authors should do the analysis and see if their assertion still stands.
4. While the authors admit to their limitations, but owning them does not absolve them of their limitations. Because only one radiologist reviewed the CMR datasets, the study had certain limitations. The addition of a second operator and a comparison of the results with inter observer coefficients would be essential.
Major points:
1. The methods section should also mention the follow-up duration.
2. Although the authors state that infiltrative disease was a criterion for exclusion (line 60), they also state (on line 178) that none of the patients included had evidence indicative of infiltrative disease. Please change.
3. Discussions should address the absence of a correlation between LGE and GLS that was identified.
4. Please delete the statement on line 244-245 (“As mentioned previously, participants in our cohort were referred for cardiac MRI mainly to exclude underlying conditions that may cause dilated cardiomyopathy and not necessarily to evaluate the prognostic significance of LGE”).
5. Even though survivors had significantly higher proBNP values at diagnosis (5573 vs. 1106), the finding that there was no difference in symptoms as assessed by NYHA class in survivors and non-survivors should be addressed in the discussion section.
Author Response
Reviewer 1
The aim of Tsabedze et al. was to evaluate the prognostic value of different CMR parameters in patients suspected to have a genetic cause of DCM. They enrolled 78 patients with idiopathic DCM who underwent CMR and found that there was a difference in terms of LVEF, LV mass index, RV volumes, mass index and RVEF between survivors and non survivors; however, none of the CMR parameters were independent predictors of all-cause mortality.
The manuscript is properly structured and well written. However, the manuscript would benefit by expanding the novel aspects and results of this study in a bit more detail.
Major points:
- As the authors point out, there are several other studies that have also looked into using CMR parameters for predicting mortality in DCM. The general statement given by the authors in lines 242 and 243, that this study has limited patient numbers is not convincing, since the number of patients in the other studies is also comparable or sometimes less than this study:
- Looi et al found increased rates of MACE in 104 patients with NIDCM and LGE (HR = 0.77, 95% CI 0.3-2.0) at 2 years (Looi JL, Edwards C, Armstrong GP, Scott A, Patel H, Hart H, Christiansen JP. Characteristics and prognostic importance of myocardial fibrosis in patients with dilated cardiomyopathy assessed by contrast-enhanced cardiac magnetic resonance imaging. Clin Med Insights Cardiol. 2010 Dec 15;4:129-34. doi: 10.4137/CMC.S5900.)
- Masci et al found that myocardial LGE is a strong predictor of worse clinical outcome in 125 patients with NIDCM (hazard ratio: 5.32, 95% confidence intervals 1.60 to 17.63, p=0.006) (Masci PG, Barison A, Aquaro GD, Pingitore A, Mariotti R, Balbarini A, Passino C, Lombardi M, Emdin M. Myocardial delayed enhancement in paucisymptomatic nonischemic dilated cardiomyopathy. Int J Cardiol. 2012 May 17;157(1):43-7. doi: 10.1016/j.ijcard.2010.11.005.)
A more detailed explanation/discussion needs to be given for the varying results/parameters reported by the other groups.
Our response: Page 10, Line 267-276; and page 11, line 276-280. We have referenced both studies. Both did not show a relationship between LGE and MACE. Page 11, line 283 – 288. We have added possible reasons for varying results.
- Major adverse cardiac events should be defined in the study in addition to all cause mortality and the frequency of hospitalizations, and the prognostic significance of LGE should be assessed in relation to both of these. This study should only be viewed as descriptive if the association between CMR parameters and adverse outcomes is not investigated in more detail.
Our response: Page 3, line 105-106. Outcomes studied have been defined.
- A power analysis of the number of patients must be performed in order to determine the effectivity and value of the conclusions made. Were there any differences regarding LGE between groups? LGE was quantified visually as the authors state in line 96-97 (“The percentage of the myocardium with LGE was calculated by counting the number of segments with LGE and dividing by 17”). However, LGE may also require quantitative analysis; thus, authors should do the analysis and see if their assertion still stands.
Our response: Page 11, line 281-283. We have emphasized the importance of quantitative parametrics on CMR Imaging. Page 11, line 323-325. The small sample size has been stated as a major limitation.
- While the authors admit to their limitations, but owning them does not absolve them of their limitations. Because only one radiologist reviewed the CMR datasets, the study had certain limitations. The addition of a second operator and a comparison of the results with inter observer coefficients would be essential.
Our response: Page 11, line 326-329. We have added “A single radiologist reported CMR imaging with no correlation by a second independent reviewer. The addition of a second radiologist and comparison of CMR imaging findings would have improved the diagnostic yield and assisted in measuring inter observer coefficients.”
Major points:
- The methods section should also mention the follow-up duration.
Our response: Page 3, line 108-109. We have added the follow-up period.
- Although the authors state that infiltrative disease was a criterion for exclusion (line 60), they also state (on line 178) that none of the patients included had evidence indicative of infiltrative disease. Please change.
Our response: sentence removed.
- Discussions should address the absence of a correlation between LGE and GLS that was identified.
Our response: page 11, line 303 -312. This has been covered in the discussion section. Global longitudinal strain (GLS) measurements are a useful prognostic marker, since it abnormalities in strain pattern are often reported, despite preserved LV function. In a study involving 15 and 33 patients with ischaemic heart disease and non-ischaemic heart disease respectively, the median GLS measurements were -15.6 (IQR: -17.9 to -11.6) and -16.0 (IQR: -19.1 to -12.7) in patients with and without LGE on imaging, respectively (p=0.212) [18]. In this study by Erley and colleagues, the global circumferential strain (GSC) had a stronger association with LGE than GLS, with an area under the receiver operating characteristic curve of 0.77 – 078 vs 0.67 – 0.72 [18]. In our study, there was no correlation between GLS generated by speckle tracking on echocardiography and the presence of LGE on CMR imaging, probably because of the limited sample size.
- Please delete the statement on line 244-245 (“As mentioned previously, participants in our cohort were referred for cardiac MRI mainly to exclude underlying conditions that may cause dilated cardiomyopathy and not necessarily to evaluate the prognostic significance of LGE”).
Our response: sentence deleted.
- Even though survivors had significantly higher proBNP values at diagnosis (5573 vs. 1106), the finding that there was no difference in symptoms as assessed by NYHA class in survivors and non-survivors should be addressed in the discussion section.
Our response: Page 11, line 289-295. N-terminal pro-brain natriuretic peptide (pro-bnp) levels are useful for the diagnosis of acute heart failure, the prognostication of patients with ischaemic and non-ischaemic cardiomyopathy and predicting the risk of cardiovascular and all-cause mortality in the general population [15, 16]. In our study, pro-bnp levels were five times higher in non-survivors, with a median of 5573 (IQR: 2471 – 8188) vs 1106 (IQR: 421 – 2826), p=0.001, suggesting that this biomarker, which is secreted in response to excessive myocyte stretch may be a useful marker for risk stratifying patients with IDCM.
Reviewer 2 Report
Idiopathic dilated cardiomyopathy (IDCM) is a well-known cause of heart failure in sub-Saharan Africa, but data were insufficient. Although there are sufficient MRI data on IDCM in the United States, Europe, and Asia, there are no data on IDCM in Africa, and this study is very significant in this regard.
Some revisions are needed, but it deserves to be published once they are made.
1)The LGE in Table 2 is a little difficult to see. The contrast between black and white needs to be adjusted, or the figure needs to be enlarged.
2)The distribution of LGE in Table 3 is also confusing. It would be easier to see the distribution if it were divided into the basal, mid, and apical regions, and what proportion of each region is represented by each of these regions.
3)I think the "Discussion" section is too short and the number of citations is too few; the discussion should include not only LGE but also Extracellular volume and T1 mapping, which could not be analyzed in this study. ECV and T1 mapping are often not available in some regions, but LGE can be used to identify cardiomyopathy in such cases.
Author Response
Reviewer 2
Idiopathic dilated cardiomyopathy (IDCM) is a well-known cause of heart failure in sub-Saharan Africa, but data were insufficient. Although there are sufficient MRI data on IDCM in the United States, Europe, and Asia, there are no data on IDCM in Africa, and this study is very significant in this regard.
Some revisions are needed, but it deserves to be published once they are made.
1)The LGE in Table 2 is a little difficult to see. The contrast between black and white needs to be adjusted, or the figure needs to be enlarged.
Our response: Figure 2 has been enlarged on page 8
2)The distribution of LGE in Table 3 is also confusing. It would be easier to see the distribution if it were divided into the basal, mid, and apical regions, and what proportion of each region is represented by each of these regions.
Our response: Figure 3 has been amended on page 9
3)I think the "Discussion" section is too short and the number of citations is too few; the discussion should include:not only LGE but also Extracellular volume and T1 mapping, which could not be analyzed in this study.
ECV and T1 mapping are often not available in some regions, but LGE can be used to identify cardiomyopathy in such cases.
Our response: we have expanded the discussion on page 10-11, and included studies by Looi and Masci, (line 267-276), a discussion on the utility of quantitative indices (line 281-288), global longitudinal strain measurements (line 303 – 312), and T1 mapping and ECV in line 313-323.